# Identification of Key Genes Related to Dormancy Control in *Prunus* Species by Meta-Analysis of RNAseq Data

**DOI:** 10.3390/plants11192469

**Published:** 2022-09-21

**Authors:** Alejandro Calle, Christopher Saski, Ana Wünsch, Jérôme Grimplet, Ksenija Gasic

**Affiliations:** 1Department of Plant and Environmental Sciences, Clemson University, 105 Collings St., Clemson, SC 29634, USA; 2Departamento de Ciencia Vegetal, Centro de Investigación y Tecnología Agroalimentaria de Aragón, Avda. Montanaña 930, 50059 Zaragoza, Spain; 3Instituto Agroalimentario de Aragón-IA2 (CITA-Universidad de Zaragoza), 50013 Zaragoza, Spain

**Keywords:** breeding, chilling and heat requirements, candidate genes, co-expression network

## Abstract

Bud dormancy is a genotype-dependent mechanism observed in *Prunus* species in which bud growth is inhibited, and the accumulation of a specific amount of chilling (endodormancy) and heat (ecodormancy) is necessary to resume growth and reach flowering. We analyzed publicly available transcriptome data from fifteen cultivars of four *Prunus* species (almond, apricot, peach, and sweet cherry) sampled at endo- and ecodormancy points to identify conserved genes and pathways associated with dormancy control in the genus. A total of 13,018 genes were differentially expressed during dormancy transitions, of which 139 and 223 were of interest because their expression profiles correlated with endo- and ecodormancy, respectively, in at least one cultivar of each species. The endodormancy-related genes comprised transcripts mainly overexpressed during chilling accumulation and were associated with abiotic stresses, cell wall modifications, and hormone regulation. The ecodormancy-related genes, upregulated after chilling fulfillment, were primarily involved in the genetic control of carbohydrate regulation, hormone biosynthesis, and pollen development. Additionally, the integrated co-expression network of differentially expressed genes in the four species showed clusters of co-expressed genes correlated to dormancy stages and genes of breeding interest overlapping with quantitative trait loci for bloom time and chilling and heat requirements.

## 1. Introduction

Bud dormancy is a critical stage in the phenology and reproductive cycle of *Prunus* species, as the survival and production of these crops depend on synchronization between growth and dormant periods with seasonal temperatures [1]. Cultivars must be adapted to temperatures in a particular growing area to avoid inadequate flowering and fruit set and therefore ensure fruit production. Thus, in *Prunus* crops, flowering occurs after a dormancy period during the autumn and winter, in which meristem growth is inactive to avoid unfavorable environmental conditions [2,3]. This dormant period, common in all *Prunus* species, is divided into two main stages [4]: endodormancy, in which bud growth is inhibited and specific amounts of chilling temperatures are required to resume growth, and ecodormancy, which refers to a period of heat accumulation needed to bud burst after chill satisfaction [5]. The transition between these stages is irreversible and is determined by the fulfillment of the chilling requirement, which is a genotype-dependent trait. This requirement is calculated by modeling temperatures in a particular environment using different approaches to estimating the chilling accumulation needed to break dormancy. The simplest model, called the ‘chill hour’ model, counts the number of hours below 7.2 °C. In contrast, a more advanced ‘dynamic model’ reports chill accumulation as chill portions that account for temperature fluctuations and is a more accurate way of estimating chill accumulation, especially in warm regions [6]. Within *Prunus* species, chilling requirement ranges from less than 100 to more than 1100 chill hours (or 4–79 chill portions) [7]. However, although both chilling and heat temperatures are necessary to reach flowering, several studies suggested chilling requirement as the major determinant of bloom time in the genus [8,9,10,11]. Nevertheless, blooming is exclusively dependent on heat accumulation in regions where the chilling requirement is not a limiting factor [12,13]. 

The fact that dormancy release, and therefore flowering time, are triggered by temperatures makes current warmer winters a critical threat for *Prunus* production in temperate regions, especially in warm growing regions such as the Mediterranean [6]. Under this scenario, several studies reported variations in the seasonal timing of bud break and flowering linked to warmer winter temperatures [9]. The early-blooming cultivars were more susceptible to frost damage, and late cultivars, which need a higher amount of chilling, had insufficient cold accumulation to break dormancy [14,15,16]. Consequently, to anticipate the effect of warmer temperatures, numerous studies focused on understanding phenology and molecular mechanisms underlying dormancy and bloom time of temperate trees. In these studies, common mechanisms between *Prunus* species were identified [17], and quantitative trait loci (QTLs), harboring major chilling requirement and bloom time genes, were mapped to overlapping orthologous genomic regions [8,10,11,18,19,20,21,22]. The *dormancy-associated MADS-box* (*DAM*) genes, with similar expression patterns during dormancy in *Prunus*, were considered key genes underlying chilling requirements in the species [23,24,25,26,27]. Additionally, numerous genomic studies have provided new information on dormancy regulation in *Prunus* tree species, mainly focusing on the transition from endodormancy to ecodormancy. At transcriptome levels, RNA sequencing (RNAseq) analyses in various *Prunus* species, including *P. persica* [28,29,30], *P. armeniaca* [30], *P. dulcis* [31], *P. avium* [32,33,34,35,36], *P. pseudocerasus* [37], *P. mume* [38,39,40], and *P. sibirica* [41], indicated that several changes occur inside flower buds at dormancy. In these studies, the number of differentially expressed genes (DEGs) differed between species, cultivars, and dormancy stages, but conserved pathways mainly related to cold acclimation [36,42,43,44], cell growth control [45], oxidative cues [46], soluble sugars regulation [40,47], and phytohormones [48,49,50] were also observed among all *Prunus* species.

Despite RNAseq revolutionizing genome-wide gene expression analysis, this technique presents some disadvantages. These include a high number of false positive rates, requiring the use of other platforms for validation, as well as the strong influence of environmental conditions, a small number of biological replicates due to sequencing cost, variability between studies as a consequence of technical differences in sample and library preparation, and single-year analyses without repetitions over seasons [51]. However, a meta-analysis of RNAseq data from different studies is a successful strategy to mitigate most of these drawbacks derived from independent studies [52,53]. Thus, to gain more detailed information about changes during endo- and ecodormancy in *Prunus* species, we performed a comparative transcriptome analysis through a meta-analysis of RNAseq data of fifteen cultivars of four *Prunus* species (apricot, almond, peach, and sweet cherry), from six independent studies, with contrasting chilling requirements and bloom time. The aim was to identify common regulatory genes associated with this critical step in the genus. In addition, an integrated co-expression network analysis of DEGs in all the species revealed the genetic coordination of genes across all species with a high correlation of dormancy stages among modules of co-expressed genes. These results also identify target breeding genes related to dormancy control within major bloom time, chilling, and heat requirement QTLs previously found in the genus, allowing breeding for increased resiliency of these crops to a warming climate.

## 2. Results

### 2.1. Summary of Meta-Analysis of RNAseq Data

In this study, transcriptomic data from 15 *Prunus* cultivars following the criteria previously described were downloaded and analyzed. These included two apricot, six sweet cherry, three almond, and four peach cultivars with contrasting chilling requirements and bloom times (Table 1). For all of them, sample points at the same dormancy stage (Endo1, Endo2, and Eco1) were selected, and a total of 136 transcriptomes corresponding to these three stages were used in analyses (Appendix A). Pairwise comparisons of these transcriptomic data between endodormancy initiation (Endo1), endodormancy progression (Endo2), and ecodormancy (Eco1) were performed to identify genes associated with these dormancy stages. Consequently, 13,018 of 47,089 annotated genes in the peach genome v2.0.a1 were differentially expressed (FC ≥ 2; *p*-value ≥ 0.05) in at least one of the analyzed cultivars (Appendix A). Of these, 3603, 4595, 6246, and 6315 genes were differentially expressed in almond, apricot, peach, and sweet cherry cultivars, respectively. The projection of these DEGs into a two-dimensional space (principal component analysis, PCA) revealed three sample clusters, each corresponding to a different species, except for almond and apricot samples that were grouped together. However, the samples were not grouped according to a dormancy stage (Appendix A).

### 2.2. Endodormancy-Related Genes

To better understand whether specific genes could be related to a particular dormancy stage, we further investigated genes showing differential expression during endodormancy. The endodormancy-related genes comprised 6860 (53% of dormancy DEGs) transcripts that were differentially expressed between the beginning and progression of endodormancy (Endo1 vs. Endo2). Endodormancy DEGs in each cultivar ranged from 472 (‘Cristobalina’) to 2375 (‘Royal Dawn’), with most of the cultivars having 650–1000 DEGs and displaying a higher proportion of upregulated genes, except the two apricots and the sweet cherry cultivar Cristobalina (Figure 1A). Of all DEGs at endodormancy, only 55, 91, 107, and 381 genes were differentially expressed in all sweet cherry, peach, almond, and apricot cultivars, respectively (Figure 1B). Even though no common DEGs were observed in all evaluated cultivars, 139 genes were of particular interest (Appendix A), as they were differentially expressed in at least one cultivar of each species (Figure 1C). GO enrichment analysis of these genes revealed terms involved in biological processes related to nitrogen compound metabolic process, external encapsulating structure organization, cell wall metabolic process, and molecular functions associated with transcription regulator activity, DNA-binding transcription factor activity, and xyloglucan:xyloglucosyl transferase activity (Figure 1D). Furthermore, hierarchical clustering of these genes revealed two main groups based on their expression profiles (clusters 1 and 2; Figure 1E and Appendix A). Cluster 1 corresponds to genes (N = 107) showing maximum expression levels at the beginning of endodormancy (Endo1) and downregulation through the progression of endodormancy (Endo2) and ecodormancy (Eco1) in almond, peach, and sweet cherry. In the two apricot cultivars, a distinctive expression profile in these genes, with maximum expression during Endo2 and Eco1, was observed (Figure 1E and Appendix A). The other cluster (cluster 2; N = 32) grouped genes only differentially expressed during Endo1 and Endo2 in sweet cherry and peach cultivars and downregulated in the four species at ecodormancy (Figure 1E and Appendix A).

To gain insight into endodormancy-related genes, we examined annotation and functional signatures of the 139 commonly DEGs in at least one cultivar of the four *Prunus* species. Genes and transcription factors primarily related to responses to abiotic stresses, cell wall modifications, and hormone regulation were observed (Appendix A). Among genes involved in responses to abiotic stresses, we especially observed overexpressed genes associated with cold tolerance, such as dehydration-responsive element-binding (Prupe.2G289500, Prupe.5G090200, Prupe.5G90500) and *MYB14* transcription factors (Prupe.3G187300 and Prupe.3G227800) that showed maximum expression at the beginning of endodormancy (Endo1). Prupe2.G139000 and Prupe.2G139500, annotated as 14kDa proline-rich protein DC2.15 and related to cold hardiness of peach, were also differentially expressed in the four species during endodormancy but, in this case, displaying maximum expression levels during the last stage of endodormancy (Endo2). Other genes related to different abiotic stress responses and annotated such as *bHLH* (Prupe.1G030500, Prupe.1G074400), *U-box* (Prupe.1G336200, Prupe.3G164200, Prupe.8G024500), or ethylene-responsive (Prupe.1G545400, Prupe.2G289600, Prupe.5G090800, Prupe.7G134100, Prupe.7G194400, Prupe.8G125100) transcription factors were overexpressed during Endo1 and downregulated at Endo2 and Eco1 (Figure 1e; Appendix A). This same expression pattern was observed for genes related to cell wall regulation, in which genes encoding for xyloglucan endotransglucosylase (Prupe.1G088900, Prupe.3G171500, Prupe.3G171600, Prupe.3G171700, Prupe.3G171800, Prupe.3G172000), glycine-rich (Prupe.2G106300), or EXORDIUM (Prupe.1G520700, Prupe.1G520800) proteins were expressed (Appendix A). 

Genes responsible for hormone signaling were also observed to have key functions during endodormancy. Several genes highly transcribed at the beginning of endodormancy in the four *Prunus* species were annotated with functions related to hormone pathways such as allene oxide synthase 1 (Prupe.1G386300), probable *9-cis-epoxycarotenoid dioxygenase* (Prupe.4G082000), *abscisic acid 8′-hydrolase* 1 (Prupe.5G013100), isoflavone reductase (Prupe.5G022800), and NDR1/HIN1-like protein (Prupe.7G097200) (Appendix A). A *dormancy-associated MADS-box* gene Prupe.1G531700 is also a member of this set of genes with a maximum expression profile at the beginning of endodormancy (Endo1) and downregulation during Endo2 and Eco1 in the four species.

### 2.3. Ecodormancy-Related Genes

The ecodormancy-related genes comprised DEGs between the transition from endodormancy (Endo2) to ecodormancy (Eco1). The 7506 DEGs (58% of dormancy DEGs) displayed a similar expression pattern, with sweet cherry ‘Kordia’ (31) and apricot ‘Palsteyn’ (2087) having the most contrasting number of DEGs in this transition (Figure 2A). However, as foreseen in endodormancy-related genes, none of the 7506 genes related to ecodormancy were differentially expressed in the fifteen cultivars at the same time, with only 16, 52, 75, and 316 genes commonly expressed in sweet cherry, peach, almond, and apricot cultivars, respectively (Figure 2B). Of these, 223 DEGs had key functions in the transition from endo- to ecodormancy in *Prunus* species, as they were differentially expressed in at least one cultivar of each species (Figure 2C). GO analysis of these genes revealed highly enriched categories associated with biological processes, including nitrogen compound metabolic, macromolecule metabolic, gametophyte and pollen development, and carbohydrate transport (Figure 2D). In addition, the hierarchical clustering of these genes highlighted two main groups based on their expression profile (Figure 2E and Appendix A). Cluster 1 corresponds to genes (N = 145) downregulated during endodormancy (Endo1 and Endo2) and showing maximum expression levels after chilling fulfillment (Eco1) in all cultivars except sweet cherry cultivars Burlat, Kordia, and Royal Dawn. Cluster 2 corresponds to genes (N = 78) downregulated during ecodormancy (Eco1) that were displaying high upregulation at the end of endodormancy (Endo2) (Figure 2E and Appendix A). 

The annotation and functional prediction of these ecodormancy-related genes revealed genetic control of cold tolerance, carbohydrate regulation, hormone biosynthesis, and pollen development (Appendix A). Among them, cold tolerance genes were the most frequent, with genes mainly grouped under cluster 2 (upregulated and downregulated during endodormancy and ecodormancy, respectively), and predicted to encode for dehydration-responsive element-binding (Prupe.2G256900; Prupe.5G090000; Prupe.5G090100), low-temperature-induced 65 kDa (Prupe.2G294400), or 14 kDa proline-rich proteins DC2.15 (Prupe.2G139000; Prupe.2G139500; Prupe.5G072800), among others (Appendix A). Several carbohydrate-related genes were also differentially expressed between endodormancy and ecodormancy. Genes annotated with catalytic activity such as NADP-dependent D-sorbitol-6-phosphate dehydrogenase (Prupe.8G083400.1), beta-fructofuranosidase CWINV1 (Prupe.3G009500), or ATP-dependent 6-phosphofructokinase (Prupe.1G444000) were downregulated during ecodormancy (cluster 2). Genes whose functions were linked to sugar transport, such as bidirectional sugar transporter (Prupe.1G220700; Prupe.3G283400; Prupe.5G146500; Prupe.8G017500) and sugar carrier (Prupe.5G083900; Prupe.5G090900), presented high transcription levels during ecodormancy (cluster 1; Appendix A). Another relevant group of genes in this stage were annotated with terms related to pathways associated with auxin (Prupe.1G503100; Prupe.8G232200), gibberellin acid (Prupe.1G313300; Prupe.4G257500; Prupe.4G080700), indole acetic acid (Prupe.8G257800), and abscisic acid control (Prupe.4G257500; Prupe.1G288000) grouped in both clusters (Appendix A). Finally, a set of six genes (Prupe.1G192200, Prupe.1G277400, Prupe.6G027000, Prupe.6G032800, Prupe.8G159600, and Prupe.8G166900) predicted to be related to pollen development and maturation were differentially expressed in these *Prunus* cultivars, showing high expression during ecodormancy (cluster 1; Appendix A).

### 2.4. Individual and Integrated Prunus Co-Expression Networks

Co-expression networks were independently constructed in almond, apricot, peach, and sweet cherry using DEGs in each species to identify genes displaying similar expression profiles during dormancy stages (Figure 3A). From WGCNA, nine modules of co-expressed genes in almond and apricot and eleven in peach and sweet cherry, containing 19 (Pdu_ME9) to 1487 genes were found (Par_ME1) (Figure 3B). Moreover, the correlation between these modules and dormancy phases revealed co-expressed genes highly associated with particular dormancy stages (Figure 3C). Modules Pdu_ME1, Pdu_ME4, Par_ME6, Ppe_ME6, Ppe_ME11, and Pav_ME8 were positively correlated to Endo1, in which GO analysis revealed significant enrichment in the regulation of the metabolic and cellular process, lipid metabolism, and cell wall regulation of co-expressed genes (Appendix A). In addition, five modules (Par_ME3, Ppe_ME3, Ppe_ME8, Pav_ME7, and Pav_ME10) showed a positive correlation with the progression of endodormancy (Endo2) in apricot, peach, and sweet cherry. However, none of the almond modules presented a statistically significant correlation to this stage (Figure 3C). Enrichment analyses of Endo2-related modules revealed genes associated with responses to abiotic stimulus and secondary metabolic processes (Appendix A). Finally, seven modules of co-expressed genes displayed positive (Pdu_ME2, Pdu_ME5, Par_ME7, Pav_ME1) and negative (Ppe_ME1, Par_ME8, Pav_ME8) correlations to ecodormancy (Eco1), with genes in positively correlated modules enriched in pathways related to carbohydrate metabolism, hormone responses, and responses to a stimulus (Appendix A). 

The independent co-expression networks for each species provided important information about co-expressed genes that might be relevant to certain dormancy stages, with various endo- and ecodormancy correlated modules sharing similar GO terms between species. Thus, to explore common expression patterns in the *Prunus* genus, we integrated the apricot, almond, peach, and sweet cherry datasets to perform a co-expression network. The 13,018 genes differentially expressed in at least one of the analyzed cultivars during the dormancy stage were considered for network construction and clustered into 25 modules of genes with similar expression profiles (Figure 4A) that contained from 122 (ME11) to 1344 (ME1) genes (Figure 4B). Of these, only ME11, which includes 122 genes, showed a significant correlation with Endo1, Endo2, and Eco1 (Figure 4C). A similar expression profile in cultivars of the four species (upregulation at Endo1 and progressively downregulation through Endo2 and Eco1; Figure 5) was observed for genes in this module. Enrichment analysis of ME11 genes did not reveal any significant GO term. An additional module, ME12, was also related to both endodormant stages (Endo1 and Endo2; Figure 4C). It contains 140 genes mainly upregulated in sweet cherry cultivars during Endo1 and further downregulated during Endo2 and Eco1, with GO terms related to response to oxidative stress, response to abiotic stimulus, and carbohydrate catabolic process (Appendix A). The ME22 was likewise correlated with Endo2 and Eco1. It displayed a common expression profile in all cultivars, downregulated during the last stage of endodormancy and upregulated during ecodormancy (Figure 5). GO analysis in this module exhibited terms mainly related to cell cycle control (Appendix A). In addition to modules that showed correlations with various dormancy stages simultaneously, we observed modules significantly associated with a particular dormancy stage (Figure 4C). This is the case for ME10, ME13, and ME21, which were positively correlated to Endo1. However, the evaluation of the expression profiles of these modules revealed genes only differentially expressed in peach, sweet cherry, and almond cultivars in each module (Figure 5). Finally, three modules (ME2, ME3, and ME14) were positively correlated with ecodormancy, consisting of genes over-expressed in this stage compared with endodormancy. ME2 and ME14 appeared to be specific to cultivars of some species (peach and apricot for ME2; apricot for ME14), and only ME3 was observed to be correlated to ecodormancy in all four *Prunus* species (Figure 5). The GO analyses of these modules revealed terms related to the nitrogen and macromolecule metabolic process (ME2), transmembrane transport, small molecule and carbohydrate metabolic process (M14), and hormone biosynthesis processes (ME3) (Appendix A).

### 2.5. Candidate Genes in Major Prunus Bloom Time and Dormancy-Related QTLs

Ten regions of the *Prunus* genome with significant bloom time, chilling, and heat requirement QTLs in almond, apricot, peach, and sweet cherry were chosen to search for candidate genes (Figure 6; Table 2). These regions were distributed along the *Prunus* genome, except chromosome (chr.) 3. Of the 134 DEGs, 49 showed annotations and functional predictions associated with dormancy control (Figure 6; Table 2 and Appendix A). Three of these genome regions, on chrs. 1 (43–43.5 Mbp), 4 (1.5–7 Mbp), and 7 (11–19 Mbp), appear to be involved in dormancy control within the *Prunus* genus, as evidenced by QTL overlap in various species, and percentages of phenotypic variance explained. Within the interval on chr. 1 (43–43.5 Mbp), in which dormancy-related QTLs were reported for almond, apricot, peach, and sweet cherry studies, three DEGs—Prupe.1G531100, Prupe.1G531600, and Prupe.1G531700, annotated as MADS-box—were found in this region (Table 2). These genes show maximum expression at the beginning of endodormancy and downregulation during ecodormancy in all cultivars (Figure 6). 

In the region on chr. 4 (1.5–7 Mbp), 7 of 26 DEGs showed annotations related to abiotic stresses such as cold resistance (Prupe.4G036800; Prupe.4G0409000; Prupe.4G046800), hormone signaling (Prupe.4G080700; Prupe.4G082000) and flowering regulation (Prupe.4G070500) (Table 2; Figure 6). Prupe.4G082000, annotated as *9-cis epoxycarotenoid dioxygenase NCED5*, appeared to be of particular interest because its expression profile was associated with endodormancy release (Figure 6). This gene was highly expressed during Endo1, downregulated during Endo2 in all cultivars but low chill ones (‘Burlat’, ‘Cristobalina’, ‘Desmayo Largueta’, ‘A209′ and ‘A340′), and later downregulated during ecodormancy in all the cultivars (Figure 6). 

Finally, in the region on chr. 7 (11–19 Mbp), in which highly significant flowering QTLs were detected for all species but sweet cherry, three of the ten DEGs were mapped with annotations related to dormancy regulation (Prupe.7G142500; Prupe.7G161100; Prupe.7G168200) (Table 2). The Prupe.7G161100, annotated as cold shock protein CS66, showed a similar expression pattern in all cultivars, overexpressed at the final stage of endodormancy (Endo2) and downregulated during ecodormancy (Figure 6). For the other QTL regions associated with particular species, detailed information on DEGs and candidate genes related to dormancy control is provided in Appendix A and Table 2, respectively.

## 3. Discussion

Recent literature on RNAseq-related studies reporting response to dormancy control in *Prunus* species has revealed a significant and unmanageable number of genes associated with genetic regulation of this period [28,29,30,31,32,33,34,35,36,37,38,39,40,41]. Most studies focused on understanding dormancy at the species level, while only two attempted to combine transcriptome data of various *Prunus* species to identify genes with similar expression patterns during dormancy transitions [30,53]. In this study, an extensive dataset comprising almond, apricot, peach, and sweet cherry cultivars was analyzed through a meta-analysis of RNAseq data to search for genes related to length and transition between endo- and ecodormancy phases. During this transition, these four *Prunus* species have conserved similar pathways, and their activation is triggered at comparable dormancy points. Our results suggest that although several pathways were activated during this period, dormancy is mainly regulated by a complex interaction of hormones, carbohydrate balance, and variations in cell wall structure, all in response to internal and external stimuli such as abiotic stresses.

In temperate fruit species, several hormones such as abscisic acid, auxins, gibberellins, ethylene, cytokinins, and jasmonates have been widely reported to control development and growth stages triggered by environmental factors [48]. Of these, abscisic acid is considered the primary hormone controlling dormancy activation and growth resumption, as its concentration was the highest at endodormancy induction and progressively decreased towards chilling accumulation in peach [58] and sweet cherry [59]. In this meta-analysis study, we found abscisic acid-related genes such as *abscisic acid 8′-hydroxylase* (Prupe.5G013100), *phosphatase 2C protein* (Prupe.5G118100 and Prupe.8G139700), and *9-cis epoxycarotenoid dioxygenases* (*NCED*; Prupe.4G082000) playing a central role on dormancy regulation in all *Prunus* species. Among these, Prupe.4G082000 had high expression at the beginning of endodormancy and progressively decreased during chilling accumulation, as previously observed in peach, sweet cherry, pear, and grapevine [33,49,58,60,61]. This gene overlapped with the main bloom time and chilling requirement QTLs on chromosome four in all *Prunus* species [8,10,11,18,19,20,22,56,62] and appears to be a key gene related to dormancy control in *Prunus* species. The Prupe.4G82000 is associated with abscisic acid accumulation [63,64,65], and together with *CYP707A* genes that are associated with abscisic acid catabolism, such as Prupe.5G013100 [61,66], regulate the balance of this hormone during dormancy. These two genes, *NCED* and *CYP707A*, had similar expression profiles in almond, peach, and sweet cherry cultivars of this study, which agrees with the hypothesis that abscisic acid catabolism is activated at the same time as metabolism to support homeostasis [67]. However, high expression of Prupe.5G013100 was observed in the two apricot cultivars included in this study, as reported for other *CYP707A* genes in some peach and Japanese pear cultivars during dormancy release [58,61,68], revealing slight differences within genotypes. In addition to abscisic acid, maintaining gibberellic acid equilibrium is another crucial process regulating dormancy [40,48]. This hormone is inversely correlated with abscisic acid, exhibiting low concentration during the early stages of endodormancy and progressively increasing until reaching the concentrations during dormancy release, as observed in sweet cherry and Japanese apricot [69,70,71]. The pattern of gibberellic acid concentration agrees with the expression profile of Prupe.4G080700.1, a *GA20x* gene associated with gibberellic acid catabolism. *GA20x* appeared essential during the transition from endodormancy to ecodormancy, as its expression was inhibited during chilling accumulation in the four *Prunus* species, as well as in rose [72] and Japanese apricot [73]. Thus, gene expression and hormone content suggest that high concentrations of abscisic acid inhibit dormancy release, while gibberellic acid promotes the transition from endodormancy to ecodormancy. Interaction of these three genes (*NCED*, *CYP707A*, and *GA20x*) is crucial to maintain an adequate hormone balance and, therefore, dormancy control since alteration in some of these genes, such as the overexpression of *GA20x*, resulted in delayed bud burst [74] and different hormone balance [75].

Flower bud protection from freezing temperatures during the winter is another critical phase during dormancy. In this study, several genes associated with cold response have been observed, mainly upregulated during the early stages of chill accumulation, ensuring flowering took place under optimal temperatures. In particular, we observed *C-repeat/dehydration-responsive element-binding factors* (*CBF/DREB*) playing a pivotal role in cold regulation, especially during the induction of endodormancy in peach, almond, and sweet cherry. In apricot cultivars, maximum expressions were reported at the end of endodormancy [30]. These genes control large parts of the cold-induced changes and have been implicated in endodormancy control in various perennial species [76]. For example, overexpression of *PpeCBF1* in apple caused a critical alteration in cold acclimation and dormancy [77]. Moreover, *CBF/DREB*s have been reported to control dormancy by binding to *DAM* genes and regulating their expression in apple, Japanese apricot, and pear [43,77,78,79]. Similarly, other transcription factors also appear to be involved in cold responses, such as *MYB14* (Prupe.3G227800), which showed maximum expression at the induction of endodormancy and was progressively suppressed during chilling accumulation, as previously reported in sweet cherry [33], *Arabidopsis* [80] and grapevine [81]. This suggests these transcription factors suppress genes subsequently activated during ecodormancy. Another cold-related gene, Prupe.2G294400, was of particular interest because of its differential expression during the transition from endodormancy to ecodormancy and overlap within a major QTL at the bottom region of chromosome 2 in sweet cherry (bloom time; [22]) and almond (heat requirements; [10]). This gene is predicted to encode a low-temperature-induced 65kDa protein, homolog to CAP160 protein in spinach that increases freezing tolerance when overexpressed [82,83,84], making it a suitable candidate associated with the phenotype variation of this QTL. In addition, two genes, Prupe.2G139000 and Prupe.2G139500, also associated with cold hardiness in peach [85], were reported here in the four species, showing maximum expression at the end of endodormancy and downregulation during deacclimation. This expression pattern is similar to that previously observed in peach and supports the hypothesis that these genes increase cold hardiness in tree shoots [85]. Another DEG, Prupe.7G161100, appeared essential in the transition from endodormancy to ecodormancy in *Prunus*. Prupe.7G161100 is a cold shock protein (CSP) co-located within the main bloom time QTL of *Prunus* species on chromosome seven [8,10,11,18,19,22] and highly expressed before dormancy release and downregulated during ecodormancy in the four species. This expression pattern correlates with that observed in *Arabidopsis*, in which distinctive expression patterns of CSP genes were associated with increasing cold tolerance [86] and negative regulation of bloom time [87]. Additionally, *AtAGL15*, a key gene controlling flowering in *Arabidopsis*, has been observed to interact with the promoter regions of two CSPs, affecting their expression [88], which supports the role of Prupe.7G161100 in the dormancy control of *Prunus* species.

In addition to hormone and cold-related genes, GO analysis revealed a large number of DEGs associated with cell wall modifications and plasmodesmata in all *Prunus* cultivars. These genes appear to be involved in critical structural mechanisms that regulate long-distance transport during dormancy onset and release through alterations in cell membranes [89,90]. For example, in this study, we found that genes involved in cell wall organization, such as those encoding for non-specific lipid transfer proteins, were upregulated during ecodormancy and facilitated lipid transport between membranes [91]. This agrees with the observation of lipid content modification throughout dormancy and its potential effect associated with cytoskeleton modification [89]. Similarly, the accumulation of callose during endodormancy establishment has been observed to be essential to block plasmodesmata and regulate dormancy [92] through the expression of *CALLOSE SYNTHASE 1*. This gene is activated by abscisic acid and represses glycosyl-hydrolase genes that, after chilling fulfillment, hydrolyze callose to resume growth [93,94]. This model was confirmed for all *Prunus* species in this study, with the upregulation of abscisic acid genes during endodormancy and the later activation of glucan endo-1,3-beta-glucosidases genes in ecodormancy that progressively degrade callose, allowing metabolite transport such as carbohydrates through flower buds [40]. These carbohydrates were essential for transition from endodormancy to ecodormancy, as an energy source, and as sugar-related signals [95,96]. In *Prunus* crops, the accumulation of soluble sugars increases during the induction of cold conditions and then declines after the chilling requirement is satisfied [59,71,97], which was also suggested as a mechanism to increase protection against low temperatures [40,98]. After chilling fulfillment, the activation of genes related to sugar transport in ecodormancy for cultivars of the four species (Prupe.1G220700, Prupe.3G283400, Prupe.5G083900, Prupe.5G090900, Prupe.5G146500 and Prupe.8G017500) was observed in this study, suggesting reserve mobilization as an essential step in resuming bud growth. 

Even though both up- and downregulated genes during endo- and ecodormancy were reported in the above discussed pathways, in this meta-analysis, we observed a large number of genes associated with flower development, whose activation was initiated during ecodormancy in the four species, and also reported in peach and apricot [30,99]. For example, it includes genes such as Prupe.1G192200 that encodes for a strictosidine synthase, homolog to *OsSTRL2* in rice, and is observed to play a key role in pollen formation and anther development [100]; the *MYB80* transcription factor Prupe.6G032800, which is essential for pollen development [101]; or the *ABORTED MICROSPORES* gene (Prupe.8G166900), whose function was associated with pollen wall formation and male sterility [102,103]. This same expression pattern, with pollen development-related genes activated after chilling satisfaction, was observed in all studied species. It is supported by microscopy observations in sweet cherry [104] and apricot [105], where stamen and pollen development were reported after endodormancy release (ecodormancy). 

## 4. Materials and Methods

### 4.1. Selection of RNAseq Studies and Sample Points Normalization

Transcriptomic studies reporting response to dormancy release in *Prunus* species were identified in the literature by searching the keywords ‘*Prunus*’, ‘dormancy’, ‘flower buds’, ‘RNAseq’, and ‘transcriptomic’ in the NCBI Sequence Read Archive (SRA) database. RNAseq datasets of these studies were selected and retrieved from NCBI SRA if they fit the following criteria: (i) RNAseq dataset was publicly available; (ii) a *Prunus* species was evaluated; (iii) transcriptome data from flower buds were sampled during dormancy stage; (iv) dormancy status of each sample point was known or could be inferred; and (v) Illumina technology was used for sequencing. Following these requirements, six studies from four *Prunus* species (almond, apricot, peach, and sweet cherry) were selected (Table 1), and transcriptomic datasets from fifteen unique cultivars were downloaded (Appendix A). For the comparative study, sample points at three common dormancy stages were identified in all cultivars according to their phenological status (Endo1, Endo2, and Eco1). Endo1 corresponds to an initial endodormancy point in which 0–20% of the chilling requirement of each cultivar was fulfilled. Similarly, Endo2 represents an endodormancy point at which more than 80% of the chilling requirement was fulfilled but heat accumulation had not been started. Finally, Eco1 refers to an ecodormancy point at which every cultivar had fulfilled its chilling requirement, but flowering had not been reached. The chilling requirements of cultivars considered in this work were calculated in each independent study by using chill hour (‘Cristobalina’, ‘Garnet’, ‘Regina’, ‘Burlat’, ‘Royal Dawn’, ‘Kordia’, ‘A209′, ‘A340′, ‘A318′, and ‘A323′) and chill portion (‘Desmayo largueta’, ‘Penta’, and ‘Tardona’) models, except for the two apricot cultivars (Palsteyn and Bergeron) whose chilling requirements were predicted based on correlations between bloom dates and chill accumulation.

### 4.2. Read Alignment and Differentially Expressed Gene Analysis

Raw transcriptomic sequences of selected studies were retrieved from NCBI SRA (https:www.ncbi.nlm.nih.gov/sra, [106]) according to their accession numbers, and sequence quality was assessed through FastQC to spot potential problems in raw sequencing datasets. Low-quality reads and adaptors from Illumina sequencing were removed using Trimmomatic software v.0.38 [107], and clean reads were mapped to the peach genome v2.0.a1 [57] using the Bowtie2 short-read aligner [108]. Peach genome v2.0.a1 was used as a reference for all species because of its assembly and annotation quality to allow a more straightforward comparison between studies. *Prunus* genus shares significant collinearity and synteny; thus, mapping sequences from other species to the peach genome is possible [109]. For each gene in a sample, transcript abundance was quantified using RSEM [110], which counts the total number of raw reads mapped per gene. Additionally, we carried out a differential expression analysis on these cultivars using edgeR [111]. Significant differentially expressed genes (DEGs) were identified with a fold change (FC) ≥ 2 and a probability of differential expression higher than 0.95. Graphical visualization of gene expression was conducted by heatmap.2 function in the gplots package implemented in R [112]. 

To allow expression pattern representation, we performed data normalization using a *z-score* for each gene using the following formula:$$z\;score=\frac{\left(TMM_{ij}-mean_{i}\right)}{SD_{i}}$$
where *TMM_ij_* = the trimmed mean of M-values (TMM) of the gene (*i*) in the sample (*j*), *mean_j_
*= TMM mean value of the gene (*i*), and *SD* = standard deviation of TMM values of all samples for the gene. For each cluster of genes showing a similar expression profile, mean *z-score* values for all considered genes within the cluster of interest were calculated for all species and dormancy points.

### 4.3. GO Enrichment and Expression Profile Analysis

Gene ontology enrichment (GO) terms were retrieved from the http://www.geneontology.org database [113,114] that provides the functions of the gene products. Enrichment analyses were conducted on GO terms for biological processes, cellular components, and molecular functions using Fisher’s exact test. Only GO terms with a false discovery rate (FDR) lower than 0.05 were considered. 

### 4.4. Gene Co-Expression Network Analysis

Weighted gene co-expression network analysis was performed using WGCNA v.1.70-3 package implemented in R [115]. Co-expression networks were independently constructed for cultivars of each species (almond, apricot, peach, and sweet cherry) and combined for all *Prunus* cultivars. Only DEGs during dormancy were included for network construction, considering a minimum module size of 10 and a merge cut height value of 0.25 as parameters. The first principal component of the expression matrix of the corresponding modules was used to define module eigengene (ME), and Pearson’s correlation coefficient at the significant level of *p* < 0.05 was used to determine correlations between modules and dormancy stages (Endo1, Endo2, and Eco1), where modules were denoted by their ME and dormancy stage by a numeric vector (‘1′ for the considered stage and ‘0′ for the others).

### 4.5. Candidate Gene Analysis

To identify candidate genes related to bloom time, chilling, and heat requirements in *Prunus* species, we searched for DEGs that co-localized within QTLs previously reported for these traits (Genome Database for Rosaceae; [116]). Thus, genome regions with overlapping major QTLs for almond, apricot, peach, and sweet cherry were investigated, and their predicted physical position in the peach genome v2.0.a1 were obtained. Genes commonly differentially expressed during dormancy in at least one cultivar of each species and mapped within these QTL regions were considered for further analysis. For these genes, literature was searched for evidence (annotation and predicted function) of involvement in dormancy control (cold response, cell wall modification, oxidative process, carbohydrate regulation, and phytohormone biosynthesis).

## 5. Conclusions

With this work, we aimed to provide a general overview of the main transcriptome changes in flower buds during dormancy in four main *Prunus* crops. We reported common pathways and genes activated at similar dormancy points in the four species through the meta-analysis of publicly available RNAseq data from fifteen almond, apricot, peach, and sweet cherry cultivars. The results revealed the presence of a similar genetic control of this dormant period in the genus, with genes associated with cold response, hormone biosynthesis, cell wall organization, and sugar content having pivotal roles in the control of dormancy. Additionally, the overlap of DEGs within previously reported main QTLs for chilling requirement and bloom time provides further support for candidate genes associated with the phenotypic variation of these QTLs. Prupe.4G082000, an *NCED* gene related to abscisic acid biosynthesis, and the *dormancy-associated MADS-box* genes located in the interval of the main bloom time QTLs on chr. 4 and 1, respectively, appear to be targeted genes for increased resiliency to climate change in *Prunus* breeding because of their correlations to endo- and ecodormancy transitions in the genus.

## Figures and Tables

**Figure 1 plants-11-02469-f001:**
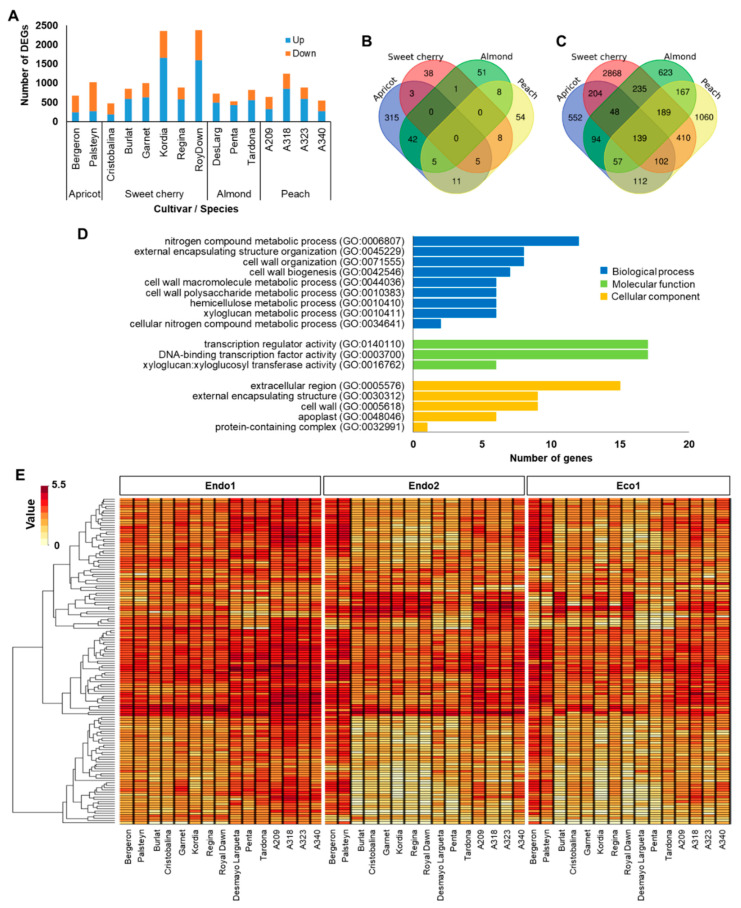
Summary of endodormancy-related genes. (**A**) Number of differentially expressed genes (DEGs) in each cultivar. (**B**) Venn diagram of common DEGs in all apricot, almond, peach, and sweet cherry cultivars. (**C**) Venn diagram of common DEGs in at least one cultivar of each species. (**D**) Enrichments in gene ontology terms for biological process, molecular function, and cellular component of common DEGs in at least one cultivar of each species. (**E**) Heatmap of the 139 commonly DEGs in at least one cultivar of each species during dormancy.

**Figure 2 plants-11-02469-f002:**
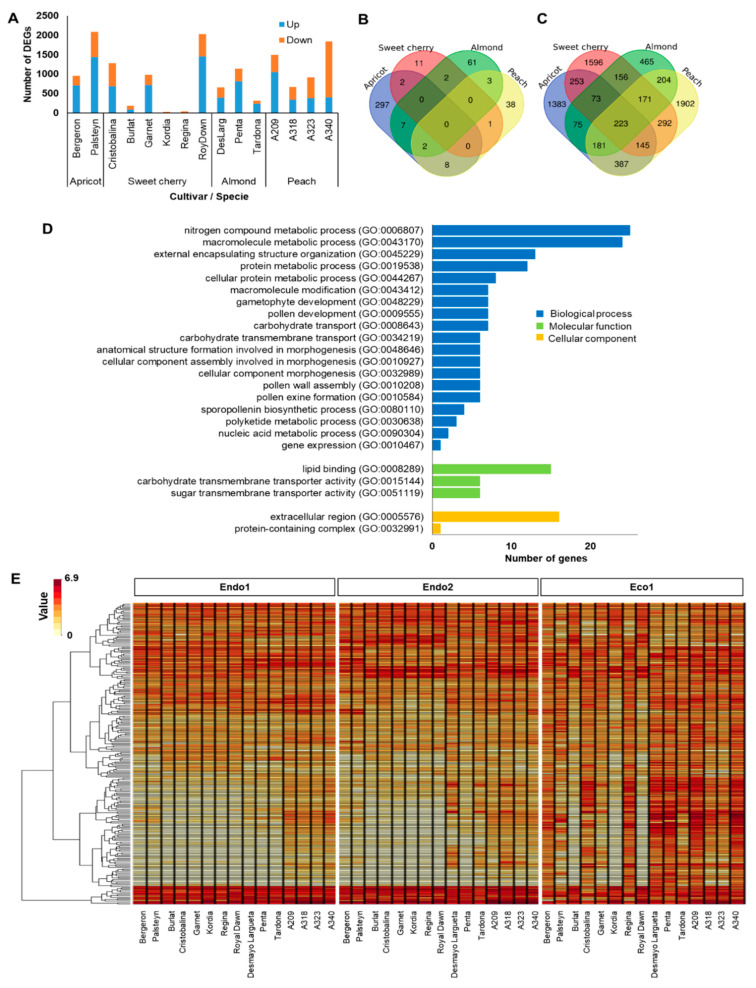
Summary of ecodormancy-related genes. (**A**) Number of differentially expressed genes (DEGs) in each cultivar. (**B**) Venn diagram of common DEGs in all apricot, almond, peach, and sweet cherry cultivars. (**C**) Venn diagram of common DEGs in at least one cultivar of each species. (**D**) Enrichments in gene ontology terms for biological process, molecular function, and cellular component of common DEGs in at least one cultivar of each species. (**E**) Heatmap of the 223 common DEGs in at least one cultivar of each species during dormancy.

**Figure 3 plants-11-02469-f003:**
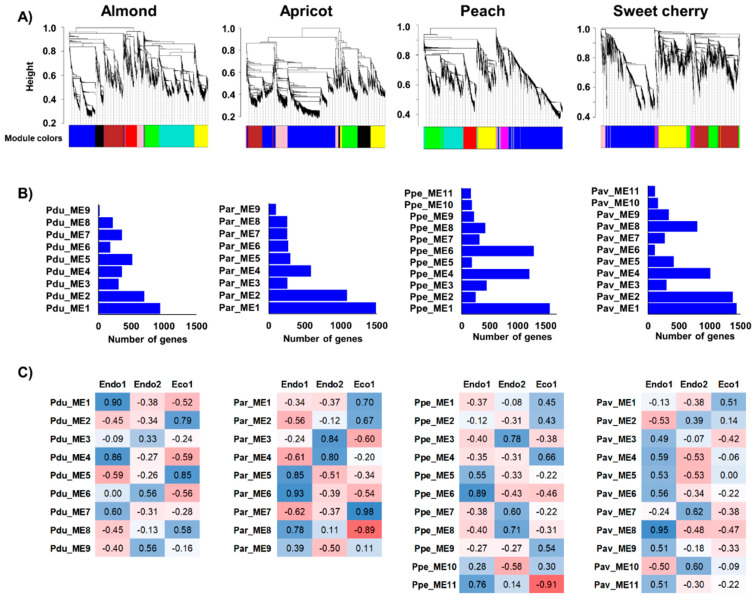
Co-expression networks constructed in almond, apricot, peach, and sweet cherry. (**A**) Dendrograms of gene clustering based on RNAseq data. (**B**) Number of genes in each module. (**C**) Correlation (*p*−values) between modules and dormancy stages (red and blue colorations indicate a negative and positive correlation, respectively).

**Figure 4 plants-11-02469-f004:**
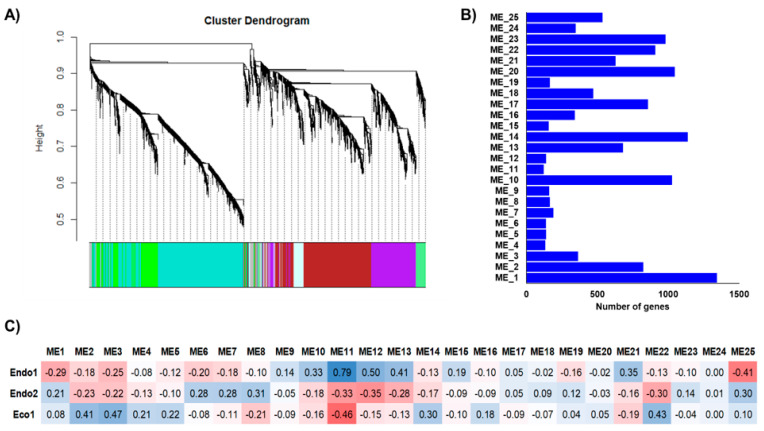
Integrated co-expression networks constructed using almond, apricot, peach, and sweet cherry differentially expressed genes. (**A**) Dendrograms of gene clustering based on RNAseq data. (**B**) Number of genes in each module. (**C**) Correlations (*p*−values) between modules and dormancy stages (red and blue colorations indicate a negative and positive correlation, respectively).

**Figure 5 plants-11-02469-f005:**
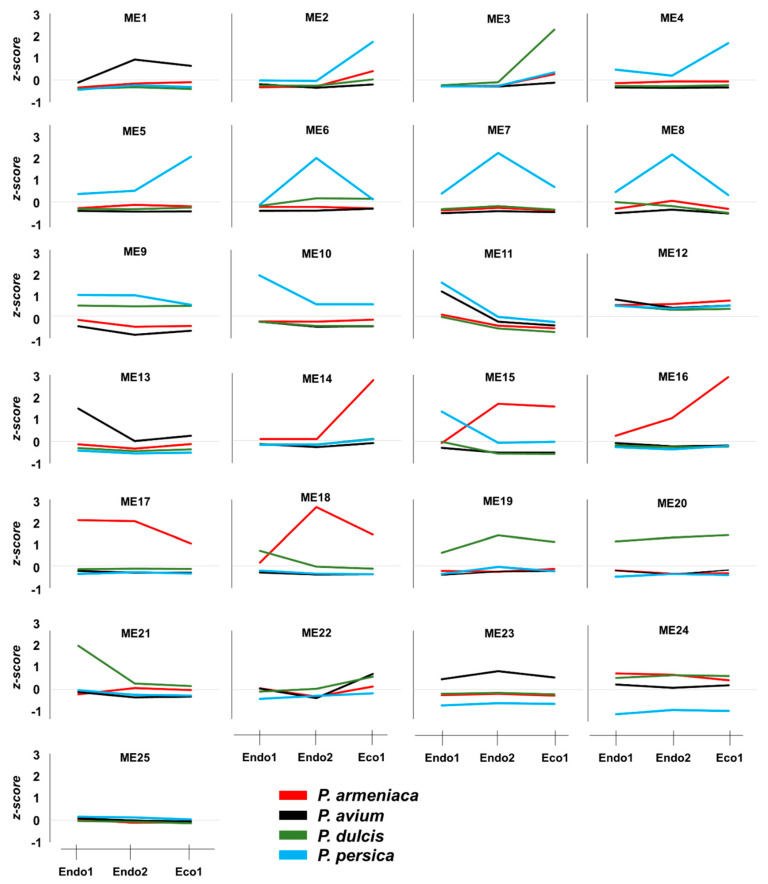
Average *z*-*scores* per species for each module from the integrated co-expression network.

**Figure 6 plants-11-02469-f006:**
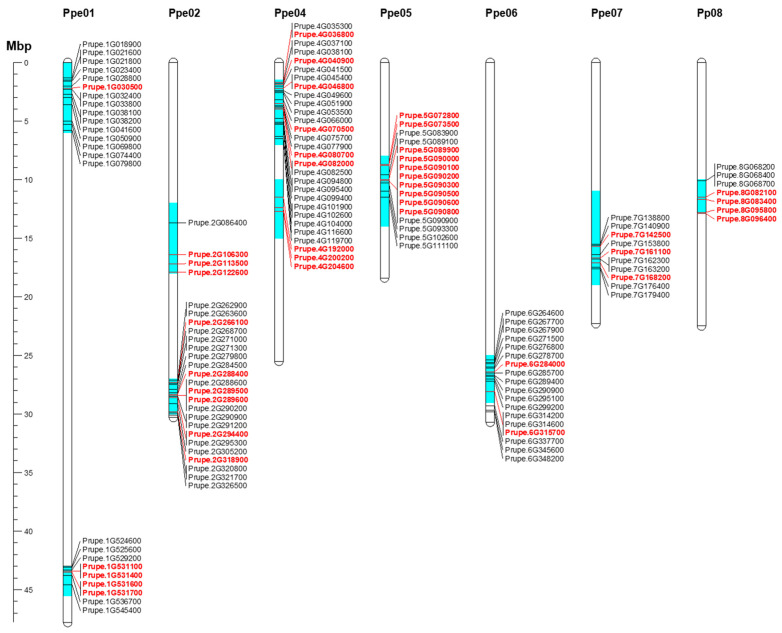
Mapping of candidate genes within major bloom time, chilling, and heat requirements QTLs on *Prunus* species [8,10,11,18,19,20,21,22,54,55,56]. Only differentially expressed genes during dormancy were considered for mapping in these QTL intervals. Genes highlighted in red show annotations related to dormancy control. Gene names were obtained from peach genome v2.0.a1 [57]. The scale represents the chromosomal position in megabase pair (Mbp).

**Table 1 plants-11-02469-t001:** Summary of metadata included in the study.

Species	Source	Cultivar	Blooming Phenotype	Location	SRA Access
*P. armeniaca*	[30]	Palsteyn	Early	Ain, France	PRJNA567655
		Bergeron	Late	Ain, France	PRJNA567655
*P. avium*	[33]	Cristobalina	Extra early	Lot-et-Garonne, France	PRJNA540235
		Garnet	Mid-season	Lot-et-Garonne, France	PRJNA540235
		Regina	Late	Lot-et-Garonne, France	PRJNA540235
	[34]	Burlat	Early	Nouvelle-Aquitaine, France	PRJNA595502
	[36]	Royal Dawn	Early	O’Higgins, Chile	PRJNA611731
		Kordia	Extra late	Valparaiso, Chile	PRJNA611733
*P. dulcis*	[31]	Desmayo largueta	Extra early	Murcia, Spain	PRJNA610711
		Penta	Late	Murcia, Spain	PRJNA610711
		Tardona	Extra late	Murcia, Spain	PRJNA610711
*P. persica*	[30]	A209	Early	Clemson (SC), US	PRJNA567655
		A340	Early	Clemson (SC), US	PRJNA567655
		A318	Late	Clemson (SC), US	PRJNA567655
		A323	Late	Clemson (SC), US	PRJNA567655

**Table 2 plants-11-02469-t002:** Genes differentially expressed in *Prunus* cultivars with annotations related to dormancy control within main bloom time, chilling and, heat requirement QTLs previously found in almond, apricot, peach, and sweet cherry. Gene ID, gene position, and QTL region based on peach genome v2.a.01.

Region	Gene ID	Position (Mbp)	Annotation
Chr1: 0–6.0 Mbp	Prupe.1G030500.1	2.2	transcription factor bHLH92
(Peach, sweet cherry)	Prupe.1G069800.1	5.0	probable xyloglucan endotransglucosylase/hydrolase protein
	Prupe.1G074400.1	5.3	transcription factor bHLH35
Chr1: 43–43.5 Mbp (Almond, apricot, peach, and sweet cherry)	Prupe.1G531100.1	43.4	MADS-box protein JOINTLESS
	Prupe.1G531600.1	43.5	MADS-box protein JOINTLESS
	Prupe.1G531700.1	43.5	MADS-box protein JOINTLESS
Chr2: 12–18 Mbp	Prupe.2G106300.3	16.4	glycine-rich protein A3
(Apricot, peach)	Prupe.2G113500.1	17.2	protein ECERIFERUM 1
	Prupe.2G122600.1	17.9	protein NRT1/PTR FAMILY 7.3
Chr2: 27–30 Mbp	Prupe.2G266100.1	27.3	methylesterase 17
(Almond, sweet cherry)	Prupe.2G288400.1	28.3	protein HOTHEAD-like
	Prupe.2G289500.1	28.4	dehydration-responsive element-binding protein 1A
	Prupe.2G289600.1	28.4	ethylene-responsive transcription factor ERF027
	Prupe.2G294400.4	28.6	low-temperature-induced 65 kDa protein
	Prupe.2G294400.2	28.6	low-temperature-induced 65 kDa protein
	Prupe.2G318900.1	29.8	two-component response regulator-like APRR5
Chr4: 1.5–7 Mbp	Prupe.4G036800.1	1.7	F-box protein At1g61340
(Almond, apricot, Peach, and sweet cherry)	Prupe.4G040900.1	1.9	NAC transcription factor 25
	Prupe.4G046800.1	2.2	CASP-like protein 1C3
	Prupe.4G070500.1	3.5	floral homeotic protein AGAMOUS
	Prupe.4G080700.1	3.9	gibberellin 2-beta-dioxygenase
	Prupe.4G082000.1	4.0	probable 9-cis-epoxycarotenoid dioxygenase NCED5
	Prupe.4G101900.1	5.2	CASP-like protein 1B1
Chr4: 10–15 Mbp	Prupe.4G192000.1	11.5	myb-related protein 308
(Peach, sweet cherry)	Prupe.4G200200.2	12.4	scarecrow-like protein 21
	Prupe.4G204600.2	12.7	gibberellin 2-beta-dioxygenase 8
	Prupe.4G204600.3	12.7	gibberellin 2-beta-dioxygenase 8
Chr5: 8–14 Mbp	Prupe.5G072800.1	8.7	14 kDa proline-rich protein DC2.15
(Apricot, peach)	Prupe.5G073500.1	8.8	4 kDa proline-rich protein DC2.15-like
	Prupe.5G083900.1	9.6	sugar carrier protein C
	Prupe.5G089900.1	10.0	dehydration-responsive element-binding protein 1A
	Prupe.5G090000.1	10.1	dehydration-responsive element-binding protein 1E
	Prupe.5G090100.1	10.1	dehydration-responsive element-binding protein 1E
	Prupe.5G090200.1	10.1	dehydration-responsive element-binding protein 1A
	Prupe.5G090300.1	10.1	ethylene-responsive transcription factor ERF027
	Prupe.5G090500.1	10.1	dehydration-responsive element-binding protein 1F
	Prupe.5G090600.1	10.1	ethylene-responsive transcription factor ERF027
	Prupe.5G090800.1	10.1	ethylene-responsive transcription factor ERF027
	Prupe.5G090900.1	10.1	sugar carrier protein C
Chr6: 25–29 Mbp	Prupe.6G284000.1	26.4	abscisic acid receptor PYL4
(Peach)	Prupe.6G315700.2	28.1	calmodulin-binding protein 60 G
Chr7: 11–19 Mbp	Prupe.7G142500.1	15.7	dof zinc finger protein DOF3.4
(Almond, apricot, peach)	Prupe.7G161100.1	16.7	cold shock protein CS66
	Prupe.7G168200.1	17.1	gibberellin-regulated protein 11
Chr8: 10–13	Prupe.8G082100.1	11.5	auxin-responsive protein SAUR50
(Peach)	Prupe.8G083400.1	11.7	NADP-dependent D-sorbitol-6-phosphate dehydrogenase
	Prupe.8G083400.2	11.7	NADP-dependent D-sorbitol-6-phosphate dehydrogenase
	Prupe.8G095800.1	12.8	protein RADIALIS-like 3
	Prupe.8G096400.1	12.9	glycine-rich RNA-binding protein-like

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
