# Peer review of "Identification of Key Genes Related to Dormancy Control in Prunus Species by Meta-Analysis of RNAseq Data"

_plants, 2022, doi:10.3390/plants11192469_

Round 1

Reviewer 1 Report

Bud dormancy is a critical stage in the life cycle of Prunus species and flowering occurs after a dormancy period during the autumn and winter, in which meristem growth is inactive to avoid adverse environmental conditions. The dormant period is common in all Prunus species and divided in two main stages, endodormancy (requirement of certain amounts of chilling) and ecodormancy (requirement of a period of heat). What are the key determinants of the bud dormancy is an important science question involving synchronized flowering and thus fruit production. Authors performed meta-analysis of RNAseq data of fifteen cultivars of four Prunus species from six independent studies to compare chilling requirements and bloom time. The results provide some important clues for understanding the bud dormancy in Prunus species.

Author Response

We thank the reviewer for taking the time to review our manuscript.

Reviewer 2 Report

In their meta-analysis, Calle et al. aimed to identify common genes that relate to dormancy control in Prunus species. The authors analyzed RNAseq data from 15 cultivars of 4 Prunus species (2 for apricot, 6 for sweet cherry, 3 for almond and 4 for peach). Although I am not an expert in the field of genetics, I could understand most parts of the article. There are some sections, however, that may require a bit of explanation in “lay” words to facilitate the reading. The results, which may be mainly confirmatory, can be useful for researchers addressing some dormancy-related questions in temperate fruit species. The manuscript seems to be well written although I do not feel confident to formally judge the writing. Below, the authors can find my comments, which should be addressed before recommending publication.

General comments:

I think the introduction can be improved by considering relevant literature in the field as well as by rephrasing some statements.

I think the conclusions can be more specific. The authors suggest that there may be a shared genetic control of dormancy within the genus. Is it also possible to point out at some differences within species? 

Specific comments

L24. There is an extra word in this line (i.e. “Introduction”).

L28-35. While I may agree with the statement “Thus, in temperate fruit tree species, flowering occurs after a dormancy period during the autumn and winter,…”, I find it weird to narrow the start of the paragraph to the “Prunus” species and then use the word “Thus” to generalize to “temperate fruit tree species”. Consider starting with the more general group of species (temperate fruit tree species) and then narrow to Prunus or to just continue with Prunus the whole introduction.

L39-42. This sentence gives the impression that only Chilling Hours are used to quantify chill accumulation. I know this model is the easiest to understand, but I would suggest to at least mention that another model (the dynamic model), which has been largely shown to perform better than the CH model, is also available.

L42-45. This may not be true if we consider a place where chilling temperatures are not the limiting factor but the accumulation of heat units. In such case, the occurrence of bloom may exclusively depend on the forcing phase. See the work by Guo et al. (2015) in Apricot trees for example.

L46-48. I understand the point raised by the authors. Nonetheless, I would say that temperate regions can be less exposed than Mediterranean climates and similar regions. In his review, Luedeling (2012) points out that cold regions (some of them temperate) may even experience an increase in accumulation of chill under future scenarios according to some chill models. Consider modifying the sentence to state the greater risk experienced by Mediterranean or warm growing regions while keeping the valid concern for temperate climate areas.

L48-52. Consider replacing the word “while” by “and”. 

L56-62. I find this sentence difficult to follow. Consider re-phrasing or splitting.

Table 1. I think the citation to Rothkegel has a typo (i.e. additional “e”). In addition, the authors can maybe squeeze the before-last column to allow the last number to fit in the last column. It does look a bit awkward now.

L101-107. Since I am not an expert in RNAseq studies, I find these lines a bit confusing. Was it the identification of transcriptomic studies a different step compared to accessing the NCBI database? If so (which is what I understood), I think the authors could explain in more detail the methodology implemented to identify these studies (databases accessed, key words used to find the studies, number of works identified in this first search, etc.). I think this may add more systematization and robustness (as in a systematic review) to the process. If this was not the case, the authors should state that the identification of studies was based on their own knowledge without applying a systematic search.

L106. I am not an English expert, but I find the word “predicted” a bit misleading. Consider modifying the word to another term such as “estimated”, “inferred”, etc.

L107-109. Since your study aims to address a genotype-dependent trait (i.e. dormancy), I think it may have been very interesting to find RNAseq datasets from different studies but evaluating the very same cultivar. Consider stating that no overlap of cultivars was found among the selected studies.

L109-115. I find this sentence particularly long and difficult to follow. Consider rephrasing it and using more periods to better explain the normalization of sample points. In addition, I think the authors may consider stating the source used to define the requirements of chill (for the definition of Endo1 and Endo2 specially). In some cases, the very same study or experiment may consider a determination of chill requirements, whereas in other cases, chill requirement values may be taken from the literature. Since some authors suggest a concerning situation regarding the method and model used to determine chilling requirements, I think it would be good to state that the authors are aware of the source of the values used to define their sample point normalization.

L117-130. I am not an expert in gene-related analysis, but this looks alright to me. The only concern I have is the use of methods/steps that only specialists could understand (e.g. “… through FastQC.”, “… using Trimmomatic software…”, “…using Bowtie2 short-reader aligner”, “… using edgeR”). Consider adding a brief explanation on what do these methodologies/steps mean in lay words. This may help non-specialists to understand what the authors did.

L130. Consider replacing “of” with “in the”. I addition, I often see a specific citation for the package (Warnes et al., 2015 in this case) and another one for the R programming environment (R Core Team).

L133 (formula). Consider using subscripts in TMM and mean. If SD is the standard deviation of TMM values of all samples for the gene “i”, it should have the subscript “i”. Is it really needed to add “– score” after the “z”. If so, consider adding it as a subscript. It does look awkward now. Also, I think the formula will look more elegant if the authors put “SDi” in the denominator instead of using “/” in the same line.

L134. Not sure I understand the meaning of “M-values”. Is it a common term that I ignore? Did I miss the explanation?

L135. I think the “j” in “meanj” should be replace by “i”?

L138. Consider adding “all” before “species”.

L140-144. Similar to the comment I made before, these lines could be extended with a brief explanation on the Gene ontology enrichment procedure.

Figure 1. Consider increasing the font size. It is difficult to read some text now. I think the direction of the cultivar name in panel “E” should be inverted as it is shown now in panel “A” (from down to up for text direction). Actually, the same axis style should be used (order and separation among species).

L214, L217. I do not see Figure 1f. Only 5 panels are shown in figure 1.

L215. Consistency between Endo1 and Endo 2 (space) as well as throughout the whole text.

Figure 2. Same as comments for Fig. 1. In addition, I think the upper part of the figure was cut. The names in the Venn diagram are not shown.

Figure 1 and 2. I don’t understand the need to include “Ecodormancy” in panel “E” of Fig. 1 and “Endodormancy” en panel “E” of Fig. 2. Why is that needed? Captions refer either to “Summary of endodormancy- or ecodormancy-related genes” in Fig. 1 and Fig. 2, respectively.

L256. Thousand separators (“,”) were used in previous lines. Make it consistent throughout the whole text.

L262. functions?

L303. Strictly speaking, “analyses” is already included in “WGCNA” (Weighted Correlation Network Analysis). Perhaps, the authors should avoid the redundancy.

Figure 3. The axis text in panel “A)” are impossible to read. I think panel identification, i.e. “A)” (in Fig. 3) versus just “A” (in Fig. 1 and Fig. 2) could be consistent. 

L307-308. Why this sentence is incomplete? It should finish with a period, I think.

L320. I think the most common use of “associated” is with the word “with” instead of “to”. Consider rephrasing.

Figure 4. I think the word “aggregated” or another explanation to clearly differentiate this analysis from Fig. 3 should be added.

L370. I think there is an additional “and” that may be eliminated. 

Figure 5. Alignment of figures is crucial when the panels share one of the axes. The panel for ME12 is clearly misplaced in relation to the panels on the left. Font size is somewhat small, making it difficult to read the numbers on the y-axis. When the text of the axis is not self-explanatory (as in the case of number on the y-axis), the axis must have a title. 

Figure 6. Is it possible to add the name of the scale next to it? Similar to an axis title.

Table 2. Since Chromosome, gene position and gene ID were all based on peach genome v2.a.01, such condition could be described in the caption of the table and removed as a differential footnote.

L427. Not sure the use of “larger” is adequate here. Please revise and correct accordingly.

L539-540. The sentence “… after chilling accumulation is satisfied hydrolyze callose to resume growth” is difficult to understand. Consider rephrasing it. 

L568. Endodormancy release?

L577-579. Not sure I fully understand how this overlap of DEGs can provide support for targeted breeding for increased resilience to climate change. Did I miss some key results related to any trait that can be of interest under expected conditions? Consider rephrasing the sentence to more explicitly state the point that the authors want to make.

L579. “Tables” in bold is an extra word.

L586. I think there is mistake. It doesn’t make sense to show “Endo1 vs Endo2” in brackets after “ecodormancy”.

L588; L591. Correlated with?

References

Guo, L., Dai, J.H., Wang, M.C., Xu, J.C., Luedeling, E., 2015. Responses of spring phenology in temperate zone trees to climate warming: a case study of apricot flowering in China. Agric. For. Meteorol. 201, 1–7.

Luedeling, E., 2012. Climate change impacts on winter chill for temperate fruit and nut production: A review. Sci. Hortic. 144, 218–229.

I do not feel confident judging the English but I thought that some expressions did look awkward in the context of the sentence, e.g.:

Use of “support a homeostasis” in L460.

Using “the transitions” when the singular form may be more appropriate in L514.

Using “focused on understanding of dormancy” did not look correct in L424.

Author Response

We appreciate the revisions by the Reviewer and thank hm/her for helping improve the quality and understanding of this manuscript. Detailed response to each comment addressing questions and suggestions is provided in the attached response to reviewer #2. Additionally, new version of the manuscript, including all reviewers' comments, was uploaded. Edits in the manuscript are highlighted in red to allow easier tacking.

Reviewer 3 Report

Review of Calle et al.

Identification of key genes related to dormancy control in Prunus species by meta-analysis of RNAseq data.

The authors sought to identify common regulatory genes in the genus associated with this critical step in Prunus species. Also performed an integrated co-expression network analysis of DEGs in all the species revealing the genetic coordination of genes across all of the species with high correlation of dormancy stages among modules of co-expressed genes.

The authors expect to extrapolate the results observed in order to explain the some outstanding characteristics of the species.

The article is generally well written, however there are a few grammatical errors and lapses in usage that could be improved.

There are also a few issues in terms of interpretation of the data, particularly in the light pervasive expression of genomes that, when addressed, would make the paper stronger.

The results section under the title " Individual and integrated Prunus co-expression networks" is too long and descriptive.

There are several phrases lacking clarity. Was the text revised by editing experts? Some of them contain important results and comments that should be better explained. Authors do not show statistical analysis for differences observed between control and treatment.

Most of the analyzed plants display a substantial difference between treatment. Were the differences statically significant? I would recommend including it in the analysis for accuracy. The processing of plants RNAs, is highly complex (see Croucher and Thomsom 2010). RNAs can be processed at the whole strand even gene level using multiple nested promoters.

This can significantly confound the interpretation of RNA-seq. Presenting information on RNA processing in plants would help the non-specialist reader. In particular direct processing of transcripts in plants is important as RNA is directly converted to protein, with control at the ribosome level.

I would rewrite it to include major results and briefly discuss the novel aspects of response brought by the study. Also, genes and pathways cited in conclusion are not the same found in the abstract. Why there is such difference?

I believe these two sections should discuss the same major achievements.

Data Availability A major flaw is that the base datasets have not been made available. As the instructions for authors state.

As the interpretation and reducibility of the study is totally reliant on the RNA-seq transcriptomic data it is not possible to reproduce the study without these datasets.

Author Response

We appreciate Reviewer #3 comments that helped improve to quality and clarity of the manuscript. Detailed response to each of the Reviewer's comment is provided int he attached response to review. Additionally, new version of the manuscript with responses to all Reviewers' comments is provided/ The edits are highlighted in red to allow easier tracking
